# Contemporary income inequality outweighs historic redlining in shaping intra-urban heat disparities in Los Angeles

Anamika Shreevastava [1,2] ✉, Glynn Hulley[1], Sai Prasanth[1], TC Chakraborty [3], Diego Ramos Aguilera [4], Kelly Twomey Sanders[4] & Yi Yin[5]

The roots of intra-urban heat disparity in the U.S. often trace back to historical discriminatory practices, such as redlining, which categorized neighborhoods by race or ethnicity. In this study, we compare the relative impacts of historic redlining and current income inequality on thermal disparities in Los Angeles. A key innovation of our work is the use of land surface temperature data from the ECOSTRESS instrument aboard the International Space Station, enabling us to capture diurnal trends in urban thermal disparities. Our findings reveal that present-day income inequality is a stronger predictor of heat burden than the legacy of redlining. Additionally, land surface temperature disparities exhibit a seasonal hysteresis effect, intensifying during extreme heat events by 5−7 °C. Sociodemographic analysis highlights that African-American and Hispanic populations in historically and economically disadvantaged areas are often the most vulnerable. Our findings suggest that while the legacy of redlining may persist, the present-day heat disparities are not necessarily an immutable inheritance, where targeted investments and interventions can pave the way for a more thermally just future for these communities.

Over the last few decades, research has unequivocally identified cities as distinctly warmer entities compared to their non-urban surroundings[1,2]. A large body of literature has also investigated the complexity of intra-urban thermal landscapes[3,4] and their associations with present-day population distributions[5–9]. Although climate change exposes rapidly growing urban populations to enhanced heat hazards[10,11], the inherent heterogeneity of cities further exacerbates the inequitable distribution of its worst impacts, especially evident in the U.S.[12,13]. For many U.S. cities, areas with poorer and predominantly non-white populations tend to be warmer[5–9]. These disparities in heat hazard have also been found to lead to unequal mortality/morbidity outcomes, making them an important policy focus for equitable urban planning[14–17]. Although the interplay of present-day socioeconomic factors, demographic characteristics, and land cover properties directly modulates the distribution of intra-urban heat exposure,

emerging evidence suggests that the roots of this thermal disparity extend beyond contemporary factors[8,18,19]. Legacy issues stemming from residential segregation policies and historical decisions based on race/ethnicity urban planning have had an enduring impact on the spatial distribution of multifaceted vulnerabilities within cities[20–22].

Discussions around historic segregation in urban areas have frequently focused on the practice of redlining, often referred to as the Home Owners Loan Corporation (HOLC) grades[23]. It emerged in the United States in the twentieth century, as part of federal housing policies, where neighborhoods were classified based on their suitability for mortgage lending[20,24,25]. Often, such policies disproportionately targeted minority communities, particularly African American and Hispanic populations[20,26,27]. Urban neighborhoods were categorized and color-coded into four different groups according to perceived desirability, with A being the most desirable and D considered 'Hazardous' and

[1]Jet Propulsion Laboratory, California Institute of Technology, California, CA, USA. [2]Environmental Science and Engineering, California Institute of Technology, California, CA, USA. [3]Pacific Northwest National Laboratory, Richland, WA, USA. [4]University of Southern California, Los Angeles, CA, USA. [5]Department of Environmental Studies, New York University, New York, NY, USA. ✉e-mail: ashreeva@caltech.edu

marked in red, hence leading to the term *redlining* (Fig. 1a). This systemic bias denied C- and D-graded neighborhoods access to loans and investments, contributing to the enduring socioeconomic disparities. This lack of financial support and equitable opportunities led to a further decline in infrastructure, amenities, and overall neighborhood conditions, including outdoor green spaces that are still visible today[28–30]. The lack of shading and evaporative cooling from trees and green spaces, combined with excess impervious built-up surfaces, increases local temperatures within these neighborhoods[31–33]. As a result, residents of historically redlined areas are often exposed to higher temperatures, and this has been observed by previous studies in multiple U.S. cities[13,29,34–39].

In this study, we focus on the Los Angeles (LA) county's metropolitan region (extent shown in Fig. 1a), which is home to nearly 25

million residents. LA' diverse topography, featuring a coastal plain rapidly rising into the San Gabriel Mountains, coupled with the strong coastal influence of the Pacific Ocean. This compressed diversity results in different Köppen climate classifications, such as Mediterranean (Csa, Csb), and semi-arid (BSh), to be found in close proximity here, which in turn leads to significant variations in temperature, precipitation, and vegetation over short distances[40]. For example, coastal areas experience a moderating marine influence with cooler temperatures and fog, while inland valleys can be significantly hotter and drier[41]. As elevation increases moving into the mountains, climate zones change dramatically. Heatwaves can occur when these coastal processes are disrupted, leading to higher temperatures[42]. Over the last few decades, LA has witnessed a gradual increase in the frequency,

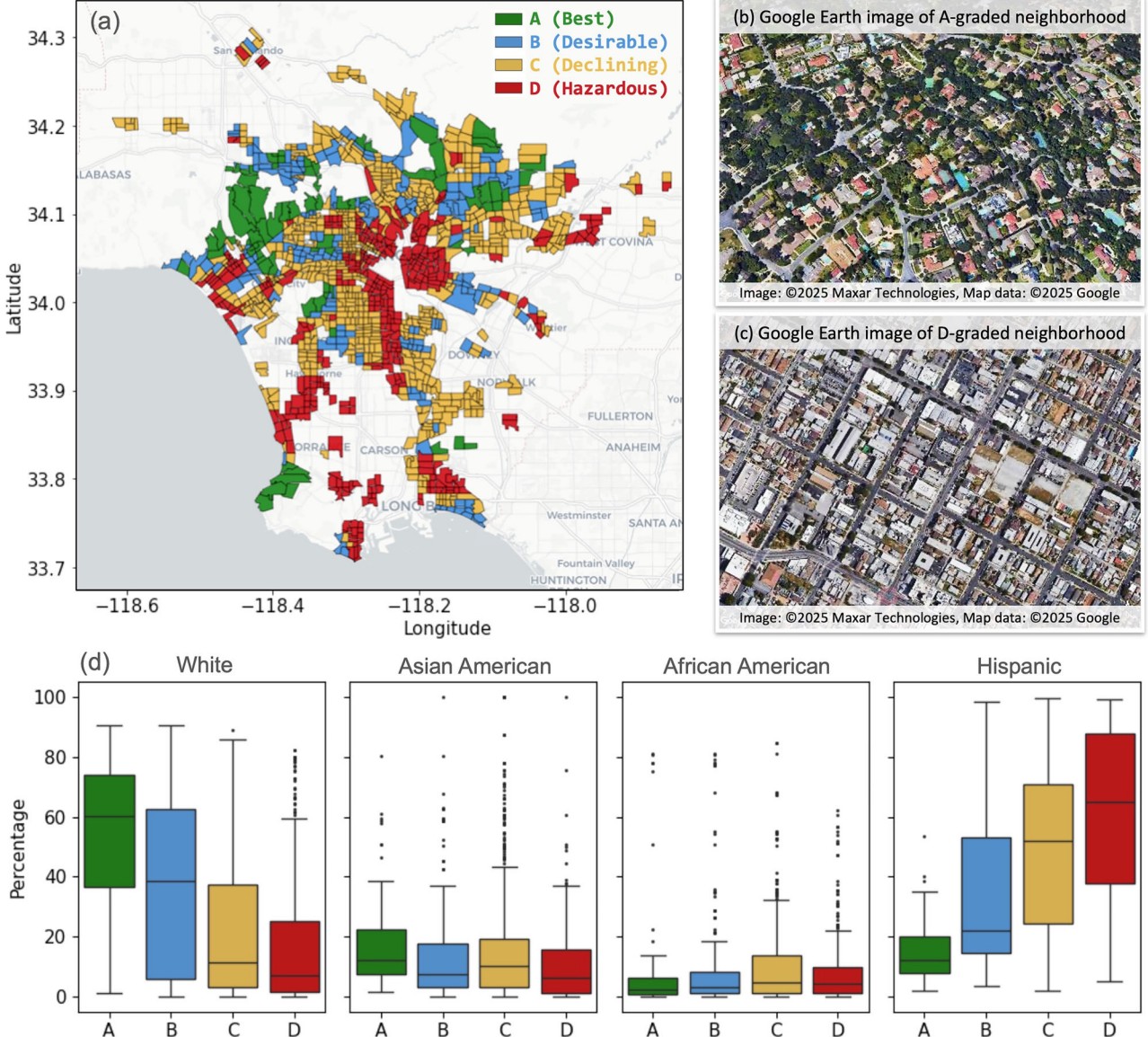

**Fig. 1 | Historical redlining in Los Angeles and its present population characteristics. a** Historic Home Owner Loan Corporation's (HOLC) map of Los Angeles where the urban neighborhoods were categorized into four grades of perceived desirability: A-graded neighborhoods (considered 'Best' for investment; marked in green), B-graded neighborhoods (considered somewhat 'Desirable'; marked in blue), C-graded neighborhoods (considered 'Declining'; marked in orange) and D-graded neighborhoods (considered 'Hazardous'; marked in red). Census tract boundaries are based on 2020 shapefiles provided by the U.S. Census Bureau. **b**, **c** Contemporary Google Earth imagery of two contrasting neighborhoods,

respectively: Pasadena, as an example of historically A-graded, and Boyle Heights, which was historically D-graded are shown. Note the substantial difference in access to green spaces between the two neighborhoods. **d** Racial and ethnic distribution in each HOLC grade based on 2020 U.S. Census tract data. Each tract is treated as an independent observational unit. The sample size of census tracts ($n$) in each of the HOLC grades are as follows: $n_A = 78$, $n_B = 195$, $n_C = 670$, and $n_D = 377$. Box plots show medians (center lines), inter-quartile ranges (boxes), 5th–95th percentiles (whiskers), and outliers (points).

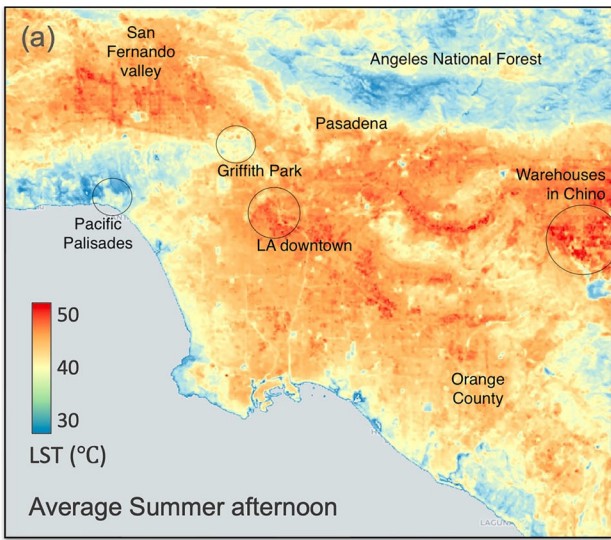

**Fig. 2 | Spatial variability of land surface temperature over the study area.** Gridded Land Surface Temperature (LST) maps of the study area are shown for the summer months (June, July, and August), with **a** representing afternoon hours (11 a.m. to 6 p.m.) and **b** representing nighttime hours (12 a.m. to 7 a.m.). The LST maps are generated using cloud-free ECOSTRESS scenes specifically from the summer months and corresponding time windows, collected between 2019 and 2023. All data were processed at a spatial resolution of 70 meters.

intensity, and persistence of heatwaves, posing a big challenge to its citizens and policymakers[43]. Additionally, LA was also subject to redlining in the past[44]. Fig. 1a, for example, displays the historically marked census tracts of LA[23] (details in **Methods**). While the discriminatory practice was outlawed in 1968 with the passage of the Fair Housing Act[45,46], its effects still linger and continue to shape patterns of inequality in many cities. Figure 1b and c, for example, illustrate how green cover is drastically different between an A- and D-grade neighborhood even today. Furthermore, significant segregation is still present across historically redlined and non-redlined neighborhoods, especially in White and Hispanic populations (Fig. 1d).

Most previous investigations into the intra-urban and intra-HOLC heat inequalities have relied on Land Surface Temperatures (LST) estimated for specific times of day from satellites in sun-synchronous orbits (e.g., Landsat at 10:00 am)[29,34–39]. Here, we leverage data from the ECOSTRESS (ECOsystem Spaceborne Thermal Radiometer Experiment on Space Station) thermal instrument onboard the International Space Station (ISS), whose precessing orbit allows observations at different times during the day, providing comprehensive temporal coverage[47,48] (see **Methods**). By leveraging ISS-borne LST observations, here, we aim to discern the diurnal and seasonal patterns that may have been overlooked in previous studies, providing a more detailed perspective on the temporal dynamics of heat inequalities and their potential implications for vulnerable and historically underserved communities.

Our exploration begins by investigating the diurnal trends in LST disparity between the two extremes of HOLC grades, D versus A, which contributes directly to the existing literature[5–9] of heat disparities due to redlining observed in various U.S. cities. Acknowledging the intricate link between income inequality and historic redlining grades[44,49,50], our paper then moves on to dissect the influence of income inequality on urban heat within each HOLC grade. Specifically, we aim to unravel the relative importance of present-day income inequalities versus century-old segregation practices. As part of this analysis, we delve into the areas that used to be historically redlined but are currently affluent. We then extend our investigation to study seasonal patterns and find an annual hysteresis effect wherein LST inequalities can have different magnitudes despite the same mean temperature, depending on time of year. Finally, we investigate the relative physiological heat vulnerability and socio-demographic

profiles of the populations currently residing in these neighborhoods. Through this comprehensive analysis, we aim to contribute to a more holistic understanding of intra-urban thermal disparities, emphasizing the importance of considering both income classes and historic redlining in evaluating the recovery and potential vulnerabilities within communities.

## Results

Utilizing the high-resolution standard Land Surface Temperature (LST) product from ECOSTRESS at 70-m spatial resolution, we explore distinctive urban features that manifest as hotspots at various times of the day[47,48]. For instance, during daytime hours, the large warehouses in central LA and Covina can be seen prominently due to higher rooftop temperatures, contrasting with the cooler, shaded road network within the urban canyon (Fig. 2a). Conversely, the nighttime data reveals paved surfaces such as road networks and the southern shipping dock, that retain heat from the daytime (Fig. 2b). We group the selected LST scenes into four seasons using the standard definition of seasons in the Northern hemisphere (Winter: Dec, Jan, Feb; Spring: Mar, Apr, May; Summer: Jun, Jul, Aug; Fall: Sep, Oct, Nov). The hours of day were clustered into four segments of time of day depending on how similar the spatial LST patterns were. The emerging four segments are labeled as follows: Morning (7 am to 11 am), Afternoon (11 am to 6 pm), Late evening (6 pm to 12 am), Past midnight (12 am to 7 am). The number of ECOSTRESS scenes used for each season and time segment are shown in Supplementary Fig. 1. We then compute the spatially gridded time-average of LSTs for the census tract shown in Fig. 1a for each of the sixteen combinations of season and time-of-day (Supplementary Fig. 1). This approach minimizes day-to-day variabilities or anomalies, enabling a focus on the expected value of seasonal LST signals, as illustrated in Fig. 2, and uncover the dependencies between LST and redlining disparity with respect to time of day and season.

### From dawn to dusk: Temperature distributions across HOLC grades

First, we examine the variation in LST distribution across the four HOLC grades, presented as normalized and staggered density distribution plots in Fig. 3. We discuss a typical summer day for illustration and provide the corresponding figure panels for Spring, Fall, and Winter in Supplementary Fig. 2. In the hours past midnight and before

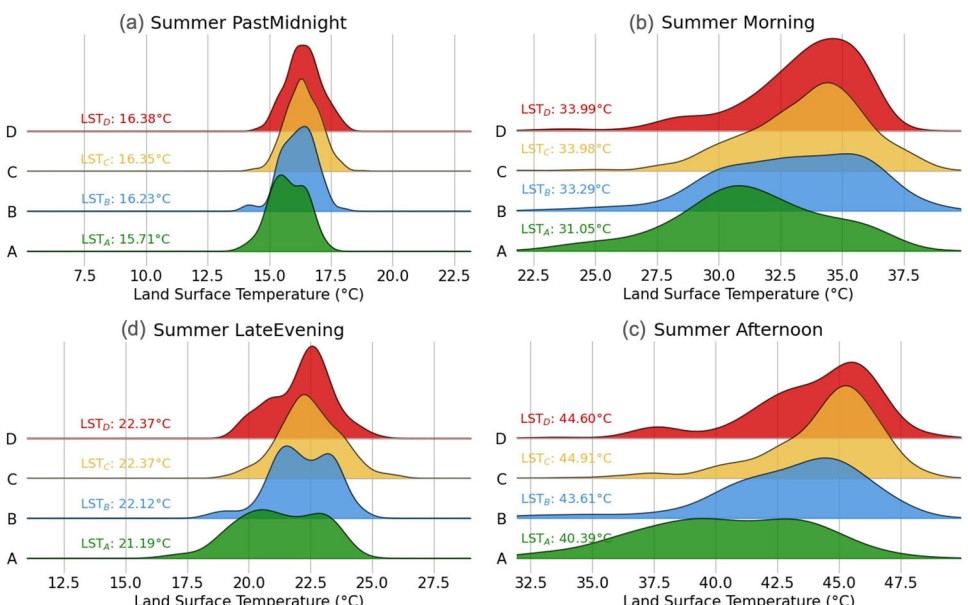

**Fig. 3 | Land surface temperature distributions by time and redlining grade.** Normalized density distribution plots of LST over the course of an average summer day (i.e., Jun, Jul, Aug) are staggered and shown for the time segments: **a** past midnight, **b** morning, **c**, late evening, and **d** afternoon. Each distribution is color- coded according to HOLC grade, and the median LST for each grade is written on the left. The horizontal axis is standardized across subplots, with each subdivision representing a consistent 2.5 °C interval to facilitate visual comparison.

sunrise, the A-graded neighborhoods experience slightly cooler temperatures ( <1 °C) while there is negligible difference in LST among the B, C, and D grades (Fig. 3a). As the sun rises, temperature disparities begin to emerge for the different redlining grades. We use the Kolmogorov-Smirnov (KS) test to confirm that these differences are statistically significant (see **Methods**). We find that C- and D-graded neighborhoods are on average 3 °C warmer than their A-graded counterparts (Fig. 3b). These disparities peak during the afternoon hours (by 4–5 °C), as mean temperatures continue to rise (Fig. 3c), before diminishing to approximately 1–2 °C during late evening hours (Fig. 3d). Notably, C- and D-graded neighborhoods retain heat longer than the greener A- and B-graded counterparts because of the increased impervious fraction. As the disparity between A versus D-graded neighborhoods in LA is primarily driven by access to green spaces, this diurnal trend mimics that of the surface urban heat island effect[51,52]. To put this in perspective, we look at the percentage of census tracts and, consequently, people residing in neighborhoods with LSTs surpassing a specified threshold, say 40 °C for instance. We find that for A-graded neighborhoods, 25% of the population live in neighborhoods with LST above 40 °C, compared to ~75% in C- and D-graded neighborhoods (Supplementary Fig. 3). This contrast impacts over 1 million additional residents during the summer in the C- and D-graded census tracts assessed in this study (Fig. 4b).

Table 1 summarizes the mean differences in LST between A- and D-graded neighborhoods ($\Delta LST_{D-A} = LST_D - LST_A$) for four time periods within each season. We find statistically significant $\Delta LST_{D-A}$ values for ~95% of the ECOSTRESS LST maps according to the KS test (Supplementary Fig. 4). The general diurnal trend remains consistent across all seasons, wherein the LST disparity is most pronounced during the afternoon hours, with summer and spring seasons displaying a $\Delta LST_{D-A}$ in the range of 3–7 °C, whereas Fall and Winter seasons have a lower $\Delta LST_{D-A}$ in the range of 2–4 °C. A detailed exploration of the drivers of seasonal differences is provided later.

### Affluence as Insulation: Joint behavior of LST, income, and historic redlining

Over the past century, the city of LA has undergone substantial transformations, including the evolution of several neighborhoods that were historically graded as C and D, as is the case of Santa Monica, now an affluent community (Fig. 4a). This transformation introduces a nuanced interplay of socio-economic inequalities involving present-day income dynamics and historical segregation practices and leads to the question: how does the inequality in LST vary as a function of present-day income inequality within these historically graded neighborhoods?

To analyze these income inequalities, we first categorize census tracts into high-income (defined as above the 75th percentile), mid-income (25th-75th percentile), and low-income (below the 25th per-centile) based on median household income (Supplementary Fig. 5). Given the evolution of present-day residential income distribution from past segregation practices[44,49,50], there is significant overlap between the low and medium income classes with HOLC grades of C and D (Fig. 4b). Consequently, almost 35% of the population (-1.6 million people) residing in historically redlined areas fall into the low-income category. Moreover, within these census tracts, nearly 40 to 80% of households fall below poverty line (Supplementary Fig. 6). Conversely, 62% of the A- and B-graded census tracts can be considered as high-income, and ~10% of these currently fall into the low-income category.

Previous research has demonstrated a negative correlation between LST and median household income across LA census tracts[53]. In this study, we juxtapose another layer of complexity from the viewpoint of historic redlining and focus primarily on the temperature distribution across income categories *within* each HOLC grade. We find that LST as a function of income differs significantly even within each HOLC grade (Fig. 4c–f). In each of the HOLC grades, the LST within the high-income neighborhoods seem to deviate the most from LST distributions in the other two income categories. During summer afternoons, for example, the higher income neighborhoods within the D-graded census tracts are 7–8 °C cooler than their lower income counterparts (Fig. 4c). Alternatively, during the coldest time of the year, winter nights, the high income neighborhoods persistently stay warmer by 1–3 °C (Fig. 4f), which is consistent with the inversion of LST-income correlation reported during wintertime in LA in other studies[53].

If we now consider the neighborhoods that are in the high-income bracket within the C- and D-graded neighborhoods, they

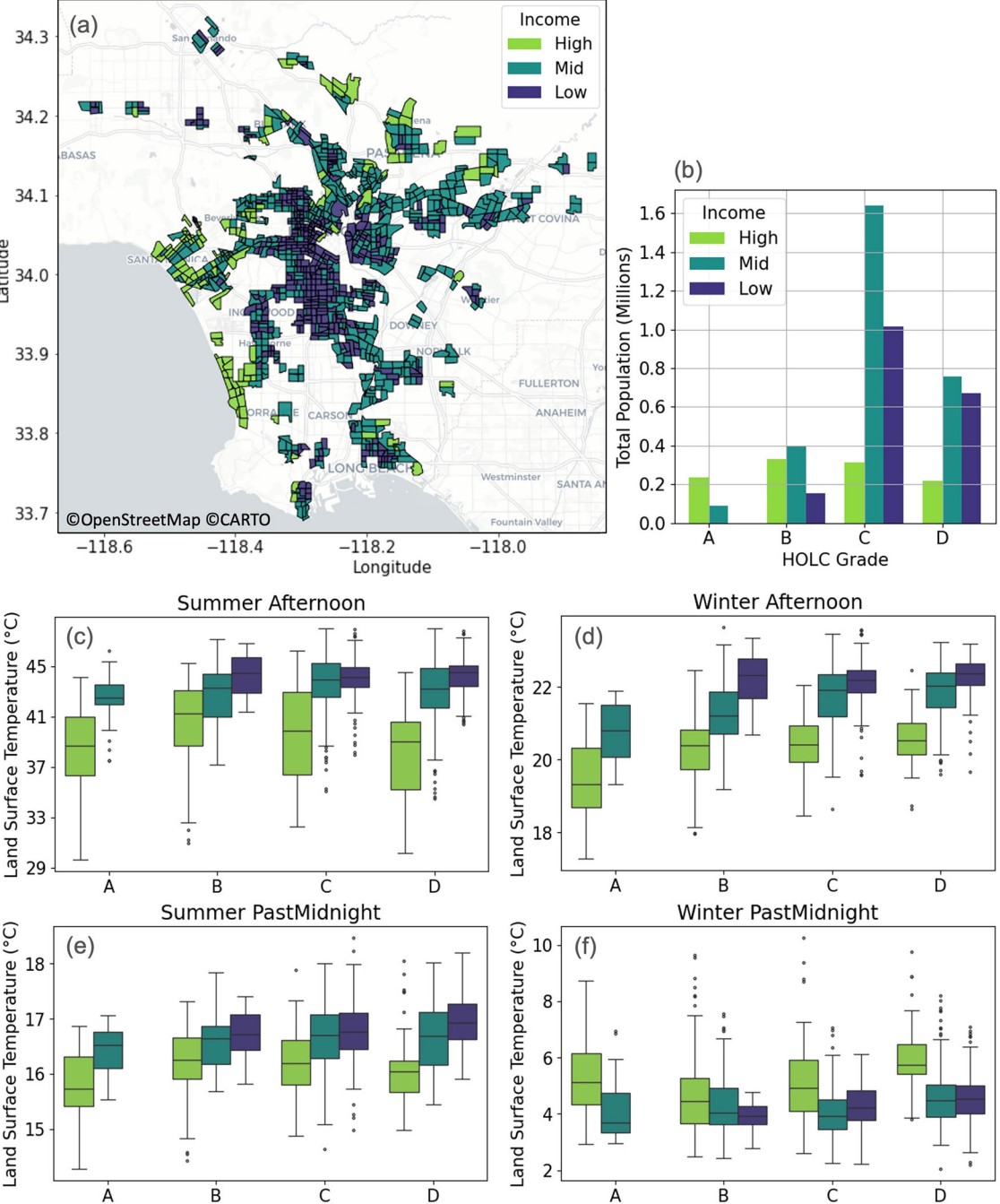

**Fig. 4 | Land surface temperature distributions by income within redlining grades. a** Present-day income distribution within historically C- and D-graded neighborhoods, with the lightest green shade highlighting areas that were historically redlined but are currently affluent. Three income classes are defined as high-income (defined as above the 75th percentile), mid-income ($25^{th}$-$75^{th}$ percentile), and low-income (below the 25th percentile) based on median household income. Census tract boundaries are based on 2020 shapefiles provided by the U.S. Census Bureau. **b** Bar graph of total population currently residing in each combination of income class and HOLC grade. **c–f** Box plots of LST distributions across the HOLC grades and income categories are shown for summer afternoons, summer nights, winter afternoons, and winter nights. All differences within each HOLC are statistically significant ($p < 0.05$). A number of census tracts ($n$) used to derive the statistics for each income group are as follows: For A: $n_{high} = 57$, $n_{mid} = 21$, $n_{low} = 0$; For B: $n_{high} = 73$, $n_{mid} = 90$, $n_{low} = 32$; For C: $n_{high} = 74$, $n_{mid} = 356$, $n_{low} = 240$; For D: $n_{high} = 45$, $n_{mid} = 172$, $n_{low} = 160$. Box plots show medians (center lines), inter-quartile ranges (boxes), $5^{th}$-$95^{th}$ percentiles (whiskers), and outliers (points).

predominantly lie in areas that benefit from the cooling effects of coastal sea-breeze influences (e.g., Santa Monica), as highlighted in Fig. 4a. Additionally, some neighborhoods around Beverly Hills, which used to be historically redlined, have also dramatically increased in affluence, possibly due to the rise of LA's Hollywood industry. This reorganization is driven by two complementary economic forces that are drivers of environmental gentrification[54]: siting[55–57] and sorting[58,59]. Simply put, the naturally cooler and greener neighborhoods increased

in affluence as they were preferentially sought out by people who could afford it[58,59]. In parallel, low-income neighborhoods are often selected as sites for industrial development, revealing disparities in siting behaviors and the impact of environmental awareness on the placement of facilities in communities of color[55–57]. While some neighborhoods serve as anecdotal examples of places that were able to invest in further greening as they became more affluent (as observed around the Hollywood region)[60], establishing a clear causal link

**Table 1 | Diurnal and seasonal variation in LST disparities ($\Delta LST_{D-A}$), defined as the difference between historically redlined (D-graded) and non-redlined (A-graded) neighborhoods ($LST_D - LST_A$) in Los Angeles**

|  | Winter | Spring | Summer | Fall |
|---|---|---|---|---|
| **Past Midnight** | −0.26 ± 0.84 °C | 0.42 ± 0.58 °C | 0.74 ± 0.34 °C | 0.33 ± 0.59 °C |
| **Morning** | 1.27 ± 1.38 °C | 1.90 ± 1.38 °C | 2.43 ± 1.04 °C | 1.04 ± 1.08 °C |
| **Afternoon** | 2.23 ± 1.01 °C | 3.87 ± 0.81 °C | 4.18 ± 0.70 °C | 2.41 ± 1.11 °C |
| **Late Evening** | 0.15 ± 0.59 °C | 1.05 ± 0.71 °C | 1.30 ± 0.53 °C | 0.52 ± 0.77 °C |

Values represent the mean temperature difference ± standard deviation across different times of day and seasons, with the greatest disparities observed during summer afternoons.

**Table 2 | Diurnal and seasonal variation in LST disparities ($\Delta LST_{HOLC,income}$), defined as difference between D-graded and low-income neighborhoods versus A-graded and high-income neighborhoods ($LST_{D,Low} - LST_{A,High}$) is shown**

|  | Winter | Spring | Summer | Fall |
|---|---|---|---|---|
| **Past Midnight** | −0.63 ± 1.00 °C | 0.47 ± 0.79 °C | 1.10 ± 0.42 °C | 0.33 ± 0.81 °C |
| **Morning** | 1.49 ± 1.86 °C | 2.75 ± 1.96 °C | 3.42 ± 1.22 °C | 1.61 ± 1.54 °C |
| **Afternoon** | 2.90 ± 1.38 °C | 5.42 ± 0.98 °C | 5.80 ± 0.73 °C | 3.31 ± 1.43 °C |
| **Late Evening** | −0.01 ± 0.75 °C | 1.43 ± 1.06 °C | 2.09 ± 0.66 °C | 0.74 ± 0.91 °C |

Values represent the mean temperature difference ± standard deviation across different times of day and seasons, with the largest disparities occurring during summer afternoons.

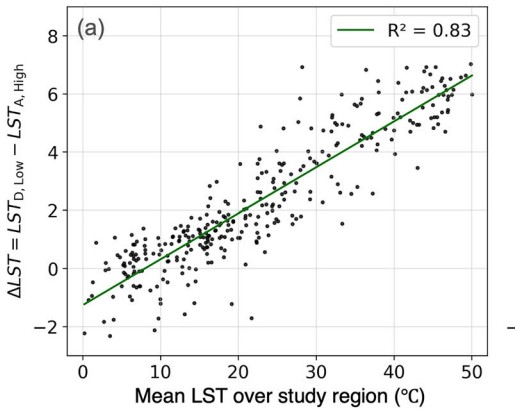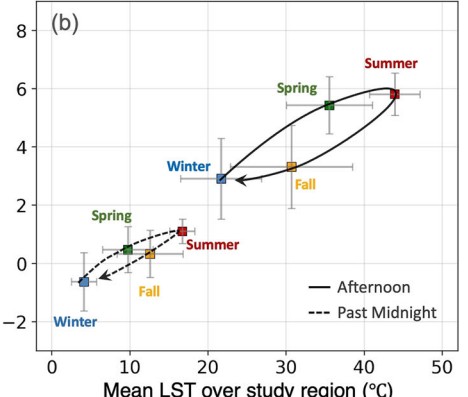

**Fig. 5 | Seasonal hysteresis in thermal disparities. a** The correlation between the mean LST disparity (defined as difference between D-graded and low-income neighborhoods versus A-graded and high-income neighborhoods, i.e., $\Delta LST_{HOLC,income} = LST_{D,Low} - LST_{A,High}$) and mean LST is shown. Each point represents a unique LST scene acquired by ECOSTRESS. The overlaid line indicates a positive correlation with an $R^2$ of 0.83 and a slope of 0.16, which implies a nearly 1 °C rise in $\Delta LST_{HOLC,income}$ for every 6 °C rise in mean LST. **b** Same as **a**, except ECOSTRESS scenes are grouped by time of day (e.g., afternoon or past midnight) and season (Winter, Spring, Summer, Fall), and the average for each group is plotted. Each point represents the mean $\Delta LST_{HOLC,income}$ for a given season and time-of-day group, with error bars showing standard deviation across scenes within that group. The seasonal trajectory is depicted using the arrows, highlighting the hysteresis for the afternoon (solid line) and nighttime (dashed line). Sample size (n = number of ECOSTRESS scenes) for each group are: Afternoon- Winter (n = 22), Spring (n = 26), Summer (n = 44), Fall (n = 33); Past Midnight- Winter (n = 18), Spring (n = 24), Summer (n = 21), Fall (n = 21).

between income, urban greening, and heat impacts[54] is beyond the scope of this study.

In summary, across the four examples shown here and those shown in Supplementary Fig. 7 for the other seasons and times of day, the LST distribution for low-income neighborhoods are not statistically different across different HOLC grades. The high and mid-income group, on the other hand, show statistically significant LST differences as a function of their HOLC grade. This is further illustrated in Supplementary Fig. 7 where we organize the LST distributions grouped by income category first, and then their HOLC grade. Therefore, we conclude that present-day income inequality is a stronger indicator of LST disparities than past redlining practices.

### Greening to senescence: hysteretic dynamics of thermal disparities across Seasons

We now introduce a more nuanced LST disparity metric using a combination of both historic redlining and present-day income

inequality, by computing the difference between the extreme cases identified above, $\Delta LST_{HOLC,income} = LST_{D,Low} - LST_{A,High}$. Compared to $\Delta LST_{D-A}$ from Table 1, we find that the general diurnal and seasonal trend of $\Delta LST_{HOLC,income}$ is still the same; but the magnitude of disparity is larger in each case, with summer afternoons' LST disparities reaching as high as 5–7 °C (Table 2, and Supplementary Fig. 4). To evaluate how LST disparity responds to extreme heat events, we evaluate the correlation between $\Delta LST_{HOLC,income}$ and mean LST for each of the 332 unique scenes collected from ECOSTRESS. Figure 5a shows a robust positive correlation ($R^2 = 0.83$) that indicates the thermal disparities are more pronounced on hotter days, such as during heatwave events, with an increase of nearly 1 °C rise in $\Delta LST_{HOLC,income}$ for every 6 °C rise in mean LST (Slope = 0.16). This demonstrates that historically D-graded and lower income communities live in neighborhoods with disproportionately higher LST compared to their historically A-graded and affluent counterparts, an effect that is more marked during a heatwave.

Shifting our focus away from the linear correlation, we find a clear hysteresis or a looping pattern when we group the data by seasons and time of day (Fig. 5b). If we consider the daily maximum temperatures that are corresponding to afternoon hours (shown in a solid line), the thermal disparity, $\Delta LST_{HOLC,income}$, increases from Winter to Summer as the mean LST increases. Additionally, although the mean LST in Spring and Fall are very similar, the respective $\Delta LST_{HOLC,income}$ are distinctly different. In essence, this means that seasonal variation of $\Delta LST_{HOLC,income}$ as a function of mean LST exhibits a time-dependent response between the forcing (i.e. mean temperature of a given day) and its effects on the system (i.e., thermal disparity, $\Delta LST_{HOLC,income}$, on the given day). We find that this hysteresis pattern is primarily driven by variations in vegetation activity, with differences in photosynthesis and evapotranspiration cooling between spring greening and fall senescence (Supplementary Fig. 8). Fall season tends to run hotter and drier in LA, with frequent heatwaves occurring later in the year causing major wildfires[43]. This is particularly notable in greener and wealthier neighborhoods where the Normalized Difference Vegetation Index (NDVI) exhibits greater annual variation than the lower income, D-graded neighborhoods with year-long higher impervious land cover fraction (Supplementary Fig. 8). The same hysteresis pattern is also evident for daily minimum temperature at nighttime (shown in dashed line in Fig. 5b) as well as other times of day which are listed in Table 2 and shown in Supplementary Fig. 9. This finding aligns with the seasonality observed in city-averaged surface urban heat islands (SUHI), with higher SUHI intensity observed in spring and summer compared to SUHI in fall and winter[61–64].

## Beyond exposure: unpacking population vulnerability to extreme heat

The overall *risk* to residents is determined by both *exposure* to heat hazards and the population's *vulnerability*, which denotes the likelihood of experiencing harm when exposed to such hazards[65,66]. Estimating heat exposure requires consideration of variables beyond surface temperature, such as near-surface air temperature ($T_{air}$, which is typically measured at 2 meters above ground level), wind speed, solar irradiance, cloudiness, and humidity[67]. Some studies have used the Weather Research Forecasting Model (WRF)[41,68] or in-situ $T_{air}$ measurements to estimate subsequent heat stress metrics[69,70]. While a comprehensive exposure assessment is beyond the scope of this study, we examined gridded $T_{air}$ from a recent global 1 km dataset[71] to compare variations in $T_{air}$ across the HOLC grades. We observe similar overarching patterns in thermal disparities in the modeled $T_{air}$ as in LST, i.e. elevated $T_{air}$ disparities are maximum during summer, especially during afternoon hours by 1–2 °C (see Supplementary Fig. 10). This finding aligns with a recent study demonstrating that LST can capture the general direction of pervasive $T_{air}$ disparities across HOLC grades although the magnitude is generally lesser for $T_{air}$ compared to LST[13]. The patterns in moist heat stress metrics, which are particularly relevant in coastal cities like LA, may be more significantly different[41,72], which requires further future investigation. Note that the modeled $T_{air}$ dataset we use is typically limited as it may underestimate actual urban thermal extremes[71] and does not capture the nuanced differences in urban form and function due to local histories such as historic redlining. Therefore, remotely sensed LST remains our best option for obtaining fine-scale spatially continuous thermal disparities within the city. Additionally, since many thermal disparities within cities arise from differences in tree cover[18], heat exposure metrics incorporating solar radiation and subsequent tree shading effects may exhibit greater disparity than seen in $T_{air}$ or moist heat stress alone[73].

We now evaluate the relative extent to which LA neighborhoods would be *vulnerable* to heat, using the IPCC's definition of vulnerability as 'the propensity or predisposition to be adversely affected, including sensitivity or susceptibility to harm and lack of capacity to cope and adapt'[74]. To this end, we evaluated the effective contribution of 26

variables that are indicative of increased vulnerability to heat impacts. These consider aspects such as preexisting health conditions[75,76], living and working conditions[77–79], financial situation[17], and social isolation of the population[80]. The source of data is discussed in the **Methods**, and a complete list of variables is available in Supplementary Fig. 11b. By design, we exclude the variables related to median household income, ethnicity, and historical redline since we evaluate heat vulnerability as a function of these factors in Fig. 6.

To obtain a single condensed metric of heat vulnerability, we performed a principal component analysis (PCA) on the identified variables[81]. Through an in-depth exploration of variable loadings, we find that the first PC explains more than one-third of the observed variance (eigenvalue: 10.0, explained variance: ~35%). PC1 captures a multidimensional sensitivity to heat with strong and positive correlations with health variables such as obesity, diabetes, and asthma, as well as moderate positive correlations with socioeconomic factors such as extreme poverty, under-education, energy deficit households, predominantly outdoor workers, and overcrowded and inadequate housing conditions. PC2 (eigenvalue: 4.0, explained variance: ~13%) has strong correlations with individuals who live alone, renters, and those lacking access to personal vehicles or the ability to drive to work, reflecting an additional sensitivity to heat driven by social isolation and harsh working conditions. Lastly, PC3 (eigenvalue: 2.5, explained variance: ~8%) captures a segment of the population that is elderly, lives alone, and experiences adverse health conditions such as diabetes and stroke (further details in Supplementary Fig. 11b). Each PC is by definition orthogonal to the others, and thus they are best treated as separate *dimensions* of vulnerability. In the remainder of this work, we focus on the first PC only, given that it is the most comprehensive, and refer to it as the vulnerability score.

The spatial distribution of the vulnerability score is shown in Fig. 6a, revealing a higher vulnerability in central LA and valley regions. Despite its cosmopolitan nature and sociodemographic diversity, LA displays stark racial and ethnic geographic segregation even today. In Fig. 6b, we plot the majority ethnic group living in each census tract (the spatial map of percentage prevalence for each ethnicity is shown in Supplementary Fig. 12). We find that the juxtaposition of heat vulnerability with the ethnic and racial distributions of the city is particularly concerning as an environmental justice issue. Notably, we find that people of color are more vulnerable to heat impacts (Supplementary Fig. 13). For example, southern and south-eastern LA neighborhoods such as Boyle Heights, Lincoln Heights, East LA, and valley regions such as San Fernando and Covina are predominantly Hispanic. On the south-central side, where heat vulnerability is high, the majority of residents are African American. In addition, these neighborhoods are also at the intersection of historic redlining (Fig. 1a), lower household income (Fig. 4a), and higher afternoon LST (Supplementary Fig. 1). Viewed from the lens of income inequality and HOLC grade in Fig. 6c, we find that the vulnerability score is low for the high-income neighborhoods regardless of their redlining history. In the medium and low-income neighborhoods, the vulnerability score scales directly as a function of HOLC grade, with the low-income and D-grade neighborhoods being the most susceptible (racial and ethnic distribution sorted by both HOLC grades and income classes are given in Supplementary Fig. 14).

Redlined areas are also shown to be linked with the heightened risk of other adverse health outcomes such as asthma and cancer, as a result of exposure to adverse environmental conditions[21,22]. This is because they are often located near industrial zones, polluted areas, or highways, leading to poor air and water quality[46,55–57]. Here, we use CalEnvironScreen's Pollution Burden[82,83] (see **Methods)** to show that D-graded neighborhoods also experience higher overall pollution levels as a combination of these factors (Fig. 6d). Despite the absence of green spaces, mitigating indoor exposure to extreme heat is still feasible through the use of air conditioning, as exemplified in cities like

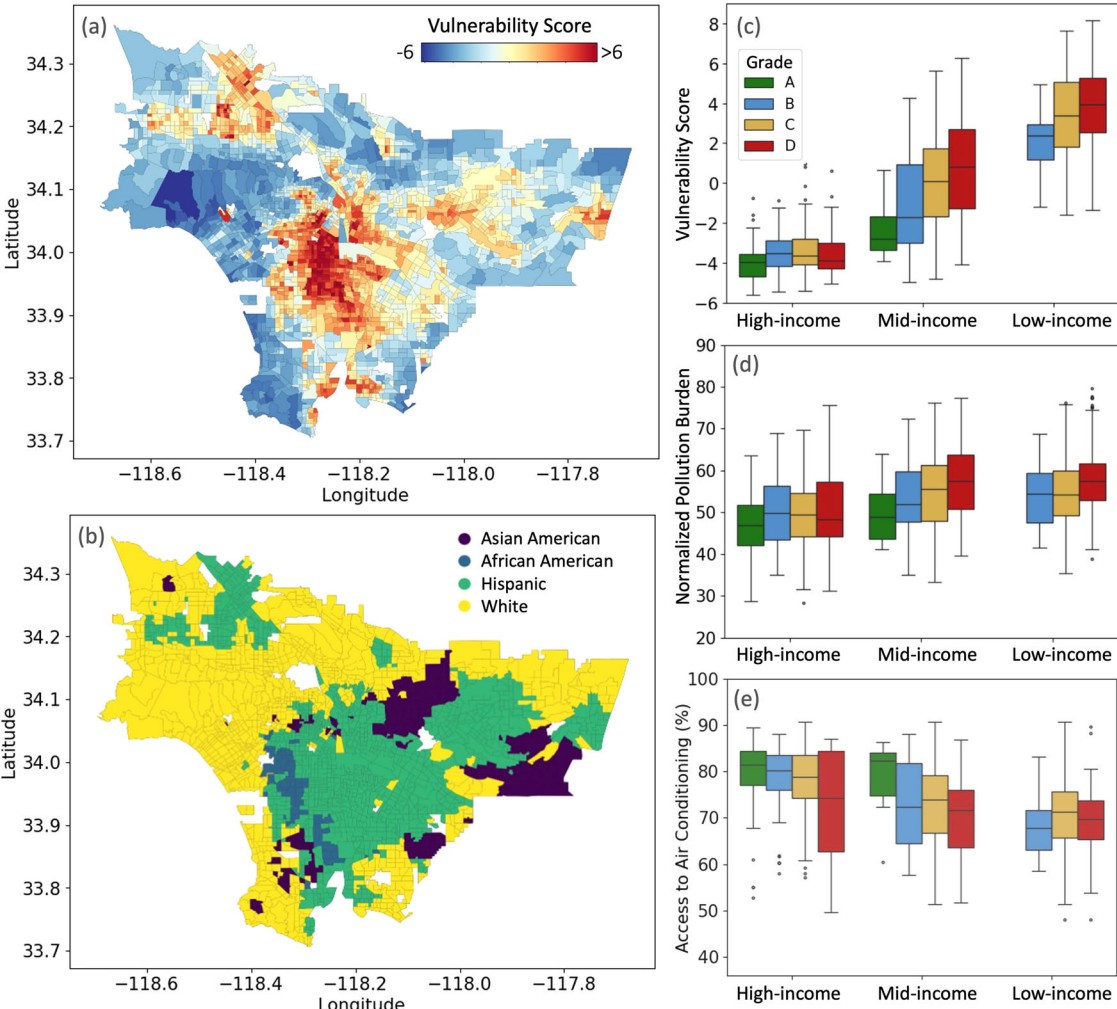

**Fig. 6 | Vulnerability to heat in Los Angeles. a** Map of the vulnerability score, computed as the first Principal Component of socioeconomic and physiological factors, is shown for all the census tracts of LA. A higher score (closer to 6) corresponds to greater vulnerability, while a lower score (closer to -6) indicates less vulnerability. **b** Map of the majority racial and ethnic groups that currently live in each census tract (Data: 2020 Census). This map serves to illustrate the racial and ethnic segregation and self-organization still prevalent in present-day LA. Boxplots of (**c**) vulnerability score (PC1), **d** normalized pollution burden score (source: CalEnviroScreen), and **e** access to air conditioning (percentage of households per census tract) organized by income classes and HOLC grades (shown in color). A number of census tracts used to derive the statistics for each group is the same as those in Fig. 4. Box plots show medians (center lines), inter-quartile ranges (boxes), 5th–95th percentiles (whiskers), and outliers (points). Census tract boundaries are based on 2020 shapefiles provided by the U.S. Census Bureau.

Dubai. Perhaps unsurprisingly, our analysis of access to air conditioning and usage shows that access to AC as well correlates strongly with income[84,85] (Fig. 6e, details in **Methods**). The AC usage only weakly correlates with HOLC grade in the middle-income category, and residents in low-income neighborhoods have limited AC usage despite HOLC grade. The widest distribution is observed in the historically D-graded high-income neighborhoods as they are mostly situated along the California coastline (Fig. 4a). The coastal cooling benefits explain why the electricity usage data reveals less AC usage, despite the affluence of these neighborhoods[86].

## Discussion

Our study sheds light on the intricate dynamics of thermal inequalities within urban areas of the greater Los Angeles region. The historical legacy of redlining, a discriminatory practice that occurred nearly a century ago, continues to cast a long shadow here, intertwining with the heightened risk of adverse heat-related outcomes for vulnerable populations. Our analysis illuminates a nuanced interplay between historic redlining and present-day income inequality, revealing both persistent challenges and potential avenues for intervention.

Comparing historically redlined areas with contemporary low-income neighborhoods uncovers persistent thermal disparities, suggesting that the injustices of the past continue to disproportionately burden marginalized communities. This raises a critical question: are the thermal disadvantages associated with these neighborhoods an irreversible consequence, inextricably woven into their physical and social fabric?

Our findings offer a cautiously optimistic answer. Lower temperatures experienced by the wealthier neighborhoods in historically redlined zones demonstrate that heat mitigation can be enhanced through resource allocation and strategic urban planning. This reorganization reflects environmental gentrification propelled by two economic drivers: siting and sorting. Affluent individuals gravitate towards naturally cooler and greener neighborhoods, driving up property values, while low-income areas often become sites for industrial development, exacerbating environmental disparities. While this finding does not diminish the urgency of addressing the plight of residents currently trapped at the nexus of historical disenfranchisement and economic hardship, it underscores that the negative impacts of extreme heat are not necessarily an immutable inheritance of

historic biases. By implementing strategic investments in green infrastructure, heat-resilient housing, and community outreach programs, we can alleviate part of the thermal burdens borne by communities historically neglected under redlining practices[87,88].

Our use of ECOSTRESS thermal data from the International Space Station provides a unique and comprehensive perspective, allowing us to explore diurnal and seasonal variations in intra-urban thermal inequities for the first time. We find that thermal disparities escalate to up to 5−7 °C during extreme heat days such as heatwaves and exhibit a seasonal hysteresis pattern driven by the spring greening and fall senescence of local vegetation. It is important to note that LA is a rather uniquely situated city with a climate shaped by complex topography and contrasting desert and coastal influences. As such, these findings may not be extendable to other U.S. cities without further investigation. Lastly, examining the demographics of the most thermally burdened and susceptible neighborhoods exposes a concerning disparity: the African-American and Hispanic populations in historically redlined and economically disadvantaged areas disproportionately suffer from this thermal injustice. Further compounding their vulnerability is limited access to air conditioning and increased exposure to higher pollution levels within their neighborhoods. Our research aims to serve as a compelling call to action, urging us to address the intertwined issues of climate change and environmental injustice. By equipping decision-makers and communities with targeted solutions, we can address some of these thermal inequities plaguing urban landscapes and build resilient pathways toward a future where heat stress does not disproportionately burden marginalized communities.

## Methods

### Home Owner's Loan Corporation (HOLC) redlining score

Redlining was a historically discriminatory practice that emerged in the United States during the 1930s and persisted for several decades. It involved the systematic denial of financial services, particularly loans and mortgages, to individuals and communities in certain neighborhoods, which was incidentally associated with income and racial/ethnic characteristics. We use Meier and Mitchell's dataset[23] for the historic HOLC map of LA, where they calculated a historic redlining score from the summed proportion of HOLC residential security grades multiplied by a weighting factor based on area within each census tract according to the 2020 census tract boundaries.

### Land surface temperature from ECOSTRESS

Our primary dataset is the standard Land Surface Temperature (LST) product from ECOSTRESS with a spatial resolution of 70 × 70 meters (at nadir) acquired over the LA region since mid-2018. ECOSTRESS, or the ECOsystem Spaceborne Thermal Radiometer Experiment on Space Station, was launched to the International Space Station in June 2018, and it currently provides the highest combined spatial (70m at nadir), spectral (3/5 bands), and temporal (3−5 days) resolution thermal infrared data from space with observations available over the diurnal cycle[47]. The ECOSTRESS LST product has been validated to a total uncertainty of 1.07 K using 14 global sites covering a wide range of surface types and atmospheric conditions[48]. We process a total of 345 complete and cloud-free scenes within the Los Angeles region based on all five years of ECOSTRESS operation (2018-2023).

### Kolmogorov-Smirnov statistic to test LST distribution dissimilarity

The Kolmogorov-Smirnov (KS) statistic is a non-parametric test used to assess the goodness of fit between an empirical distribution and a theoretical distribution or between two empirical distributions. It measures the maximum absolute difference between the cumulative distribution functions of the observed and expected data. KS statistic was chosen because the temperature distributions were non-normally distributed. The null hypothesis of the KS test is that the two distributions are identical, and a significant result indicates a rejection of this hypothesis, suggesting a significant difference exists between the distributions being compared.

### Principal Component Analysis of vulnerability to heat

The social, demographic, and health data employed in the vulnerability index is obtained from four publicly available sources at the census tract level: the American Community Survey (ACS), CalEnviroScreen 4.0 (CES)[82], the Climate and Economic Justice Screening Tool (CEJST), and CDC's 500 Cities Project (CDC), encompassing variables measuring living conditions, social isolation, age and health, literacy, occupation, and transportation barriers that have been demonstrated to enhance the relative risk of heat mortality or morbidity[17,75–80]. We perform feature selection to remove redundant variables using a hierarchical clustering approach, eliminating variables that have correlations > 0.9, resulting in 26 variables being selected. We also perform feature scaling before starting the PCA. We run the PCA using the Python *scikit-learn* PCA package, which returns the relative influence of each variable on the principal components. Please see Supplementary Fig. 11 for further details.

### CalEnvironScreen's pollution burden score

CalEnviroScreen is a tool developed by the California Environmental Protection Agency to assess and identify communities disproportionately burdened by pollution[82,83]. It calculates pollution burdens by considering various factors such as air quality (ozone, particulate matter), toxic releases, impaired water bodies, solid waste sites, cleanup sites, traffic density, and pesticide use. It aims to address environmental justice concerns by identifying and assisting disadvantaged communities vulnerable to the adverse effects of pollution.

### Estimates of air conditioning access and usage

The estimates of air conditioning prevalence at the census tract level were derived from various demographic factors, climate conditions, and geographic differences[84]. The logistic model was constructed using American Housing Survey metropolitan and national microdata spanning from 2013 to 2019 and was then applied to American Community Survey data to predict air conditioning prevalence for 115 metropolitan areas in the United States. This is closely related to the probability of AC ownership. In most regions of the U.S. where AC penetrations tend to be very high, this correlation is acceptable, but California has lower penetrations than most US states because of the coastal influence, and this relationship breaks down[86]. We improve upon this estimate by assessing the electricity-temperature relationship of smart-meter records of 200,000 homes in Southern California over the years 2015-2016, using data provided by local utility Southern California Edison (SCE)[85]. As a result, our estimates more closely capture the actual usage of AC. Please see Supplementary Fig. 15 for further details.

### Reporting summary

Further information on research design is available in the Nature Portfolio Reporting Summary linked to this article.

## Data availability

The ECOSTRESS Land Surface Temperature (LST) data generated in this study are freely available at the NASA LP DAAC repository under accession code ECO2LSTE v001. The smart meter data used to estimate air conditioning usage is protected by an NDA and not available due to privacy agreements with Southern California Edison (SCE). The sociodemographic, environmental, and health variables used for vulnerability assessment are publicly available from the American Community Survey (ACS), CalEnviroScreen 4.0, CDC 500 Cities Project, and the Climate and Economic Justice Screening Tool.

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

## Acknowledgements

This research was carried out at the Jet Propulsion Laboratory, California Institute of Technology, Pasadena, CA, USA, under contract with the National Aeronautics and Space Administration (NASA). A.S. and Y.Y. were supported by the NASA ECOSTRESS Science Team Project (grant number 80NSSC20K0078). G.H. was supported by NASA's ECOSTRESS mission funding. D.R.A. and K.T.S. were funded by the National Science Foundation (grant numbers NSF CBET-1752522 and NSF CBET-1845931). T.C.'s contribution was supported by a NASA Interdisciplinary Research in Earth Science grant (grant number 80NSSC24K0505). The Pacific Northwest National Laboratory is operated for the U.S. Department of Energy (DOE) by Battelle Memorial Institute under contract DE-AC05-76RL01830.

## Author contributions

A.S., G.H., S.P., and Y.Y. contributed to the conceptualization of study, design of experiment, and analysis of all data. T.C. contributed to the data collection and analysis of thermal data. D.R.A. and K.T.S. provided data for access to air-conditioning, and contributed to the data collection, processing, and the principal component analysis. A.S. wrote the manuscript and all other authors provided critical feedback and input. Y.Y. secured the main funding source for this research.

## Competing interests

The authors declare no competing interests.
