## [Transparent Peer Review file · Nature Communications]

Contemporary income inequality outweighs historic redlining in shaping intra-urban heat disparities in Los Angeles

Corresponding Author: Dr Anamika Shreevastava

Version 0:

Reviewer comments:

Reviewer #1

(Remarks to the Author)

This is a worthwhile study of Los Angeles (not clear in the title that it is a single city) about intra-urban heat island exposures. The abstract describes its major contributions accurately: 1) compares temperature differences among areas that were redlined a century ago using HOLC maps and present-day income inequality; 2) uses superior LST data compared to older studies that allow better diurnal and seasonal comparisons of temperature differences. The 4x4 grid study design of season and times of day/night is systematic, the analysis supports the finding that present-day income is more influential than redlining in predicting differences in heat exposure, and the writing is very clear.

The manuscript should review and acknowledge previous research on intra-urban heat temperature disparities among socioeconomically diverse and racially/ethnically segregated neighborhoods. The second sentence of the Introduction (lines 3- 4) is very misleading. Studies of intra-urban landscapes and inequities in extreme heat impacts are voluminous dating back to the early 2000s (e.g., US cities Phoenix, Philadelphia, Detroit, Chicago, Atlanta, New York, and others, as well as cities in Asia and Europe). Many US cities have developed heat vulnerability maps and action plans for mitigating heat effects in lower-income neighborhoods. Federal agencies such as the EPA Heat Islands and Equity, the CDC and NIH Heat Equity have long promoted urban heat/health equity to address problems in economically disadvantaged and racially segregated neighborhoods. In line 26, the authors cite several redlining studies on heat, but again it is not acceptable to ignore the large body of research on contemporary inequalities that underlie both the concept and methods of their study, including in their own city, Los Angeles, as well as very similar findings about neighborhood inequality. Some of the prior research shows differential impacts on heat mortality and morbidity, which this study does not do.

The manuscript would benefit from a brief section on LA climate, vegetation, and socioeconomics to establish a relevant context for the study. It is a rather unique combination of desert and coastal characteristics that differ from most other large US cities, so generalizability to other places might be questioned. The map figures are not adequately marked to give readers a sense of the expanse or "neighborhood" composition of the urban area. Sometimes the authors give interesting examples of particular places, but they are not easy to see on the maps. Figure 1 or Figure 2 might be a good place to orient readers with more prominent indications of places referred to in the text. Figure 4 is very hard to follow along with the text because it is attempting to depict relationships among three or four important variables. I do not see any red circles in Figure 4a. I recommend reviewing this section with clarity in mind.

The definition of hysteresis in lines 164-166 is conveyed in technical terms that obscure the important substantive finding. The authors would do well to explain the specific process in a more straightforward way.

ECOSTRESS is a welcome advance in measuring land surface temperatures. As the authors say in line 162, it is necessary to consider some other weather variables to estimate heat stress on humans. Tair at a 1 km scale is one way to do this, but it is still a rather crude estimate of microclimates. It should be acknowledged that other studies have used the Weather Research Forecasting Model (WRF) and even individually-experienced temperature sensors to improve upon air temperature measurements because air is more closely related to human stress than LST.

It is standard to use PCA to construct heat vulnerability indexes. The authors have included some atypical variables in their analysis, which is good, but they don't explain or label the factors. If I understand correctly, only the first factor that explains 35% of the variance is used in the vulnerability index. More discussion of what this analysis shows about the social sources

of heat vulnerability would be helpful. I don't see an explanation for the choice to omit the other two factors that increase the explained variance in the data. Also, please give clear color guidance in Supplemental Figure 11 for the significance and direction of factor loadings.

The discussion of "access" to air conditioning in lines 231-238 is somewhat confusing. Does access mean having the equipment in the home, choosing to use it or not, or not using it because it is unaffordable? Considering other research on indoor heat deaths and social surveys of customers, it is likely that residents in low-income areas don't use air conditioning because they can't afford it. In Supplemental Figures 11 and 15, it looks like low usage areas are in downtown LA where African Americans and Latinos live.

This brings me to my final comment that refers to Supplemental Figures 13 and 14. These figures show that African Americans are as highly exposed to heat risks as Hispanics in LA. I believe the data indicates that lines 266-267 in the Discussion should be amended accordingly.

Reviewer #2

(Remarks to the Author)

This article is an extremely important contribution to the literature about urban heat islands. It elucidates several important drivers of inequitable vulnerabilities to climate change, in particular dissecting the relationship between historical redlining and current economic and racial disparities. The outcomes are well-articulated in the introduction and discussion. It should be published ASAP.

The robust statistical analysis and referencing of prior research and source datasets are exceptional. While I do not conduct this type of quantitative analysis in my own research, with a bit of effort, I could follow the analysis, with two exceptions, as explained below.

First, the explanation of how the first PC was used to create the vulnerability score should be strengthened. What exactly does the scale of -6 to 6 represent? And what does the red and blue color coding represent? The first question should be answered more clearly at the end of the second paragraph in Section 2.4. The second question could be answered more clearly in the caption in Supplementary Figure 11b.

Second, a more minor addition I recommend, also in Section 2.4, is to explain the difference between LST and near-surface air temperature. In particular, the only place I could find an indication of the height above ground at which the air temperature was taken is in the caption for Supplementary Figure 10. It would be helpful to readers if this distance were noted in the description of the dataset in the narrative in Section 2.4.

Finally, I noted just a couple of line edits, as follows:

- Line 32, sentence starting "Further..." needs editing for clarity, as it appears to be an incomplete sentence
- Line 190, "maybe" should be "may be"

Reviewer #3

(Remarks to the Author)

The study aims to disentangle the effects of historic redlining and current income inequality on urban heat disparities. The results indicate that present-day income inequality is a stronger indicator of heat burden than historic redlining. While the research addresses an important topic—intra-urban inequities—there are several critical issues that need to be addressed to strengthen the manuscript.

Major comments:

1. This study focuses solely on one city, Los Angeles. To attract more attention, the narrative should be framed from a broader perspective. For example, it could include major cities across the United States or even globally. The implication of this study is limited to a single city.
2. Lines 3-4: Previous studies have extensively analyzed intra-urban inequalities. It is necessary to further expand the background information and include relevant citations. Additionally, it is important to highlight the limitations of the previous studies and the innovative aspects of this research in the Abstract and Introduction.
3. Lines 38-41 and Lines 283-292: What is the spatial coverage of ECOSTRESS observations? Can it be used for global or U.S.-wide studies? Why was Los Angeles chosen as the study subject? Moreover, this study focuses only on cloud-free land surface temperatures. Considering the influence of clouds on temperature could impact the study results. For instance, higher temperatures promote precipitation and cloud cover, thus reducing incoming solar radiation and mitigating urban warming. This limitation should be acknowledged in the article.
4. Land surface temperature does not accurately reflect human thermal sensation. It is necessary to supplement the study with surface air temperature data from meteorological stations. Moreover, the temporal resolution and frequency of the data are not clearly described in the manuscript. It is not clear what data has been used to derive the average LSTs.

Minor comments:

1. Line 83 and Lines 102-103: In addition to seasonal differences, the reasons for daytime and nighttime variations should also be specifically discussed and analyzed.
2. Lines 127-129: During winter, why are temperatures higher in high-income areas compared to low-income areas? Is it due to the effect of anthropogenic heat?
3. Lines 263-264: How is a heatwave event defined in the study?
4. The research indicates that present-day income inequality is a stronger indicator of heat burden than historic redlining. This content is not reflected in the title.
5. The abstract should briefly introduce the background of historic redlining. Or at the beginning of the introduction.

Version 1:

Reviewer comments:

Reviewer #1

(Remarks to the Author)

Thank you to the authors for addressing the reviewers' commentary. Their efforts have improved the manuscript. Reviewer 1's reactions to revisions made in response to my comments:

1. Thank you for removing the misleading sentence in the first paragraph. Although the number of citations added does not fully reflect the large literature on intra-urban heat inequalities, it makes the necessary point for this manuscript.
2. The new description of the LA climate and geography is not exactly what I had in mind. The complexities of the Southern California climate zones might be better appreciated by referring to the unusual close proximity of Koppen classifications; i.e., something like lines 278-279 would be beneficial in the introduction. Regarding comments about the locations of places on the maps, a larger point is that we should have an exact definition of the boundaries of the study area. The "Los Angeles region" is too vague – is it an MSA or some other construction that you are using? The use of terms like city and neighborhoods throughout the text deserves some clearer context because there are many cities in this region. Figure 2 is an example – for readers who are not familiar with SoCal, there is no orientation to where the city of LA is, let alone central LA, urban canyons, and shipping dock. Is a popout map an option? Figure 3 gives some markers about where places are located.
- 3, 4. Thank you for the clarifications.
5. I might have made a different decision about how to use the PCA. Collectively, I think we are learning this method is not the best way to differentiate the different types of human vulnerability. Factor 1 points to a broad constellation of vulnerable individuals but it is so general that it would not provide much guidance on protections. Nevertheless, that is not the point of this study and we don't have anything better as a summary measure.
- 6, 7. Thank you for the clarifications.

Reviewer #2

(Remarks to the Author)

The authors have improved their already-strong manuscript in their responses to the first round of reviewer comments, resulting in an excellent article that I look forward to seeing in publication. I have no further comments.

Response to reviewers for manuscript: NCOMMS-24-14163

Unraveling the role of historic redlining and present-day inequities on intra-urban heat disparity

New Title: Present-day inequities in Los Angeles have a stronger role on intra-urban heat disparities than historic redlining

Reviewer #1:

This is a worthwhile study of Los Angeles (not clear in the title that it is a single city) about intra-urban heat island exposures. The abstract describes its major contributions accurately: 1) compares temperature differences among areas that were redlined a century ago using HOLC maps and present-day income inequality; 2) uses superior LST data compared to older studies that allow better diurnal and seasonal comparisons of temperature differences. The 4x4 grid study design of season and times of day/night is systematic, the analysis supports the finding that present-day income is more influential than redlining in predicting differences in heat exposure, and the writing is very clear.

1. The manuscript should review and acknowledge previous research on intra-urban heat temperature disparities among socioeconomically diverse and racially/ethnically segregated neighborhoods. The second sentence of the Introduction (lines 3- 4) is very misleading. Studies of intra-urban landscapes and inequities in extreme heat impacts are voluminous dating back to the early 2000s (e.g., US cities Phoenix, Philadelphia, Detroit, Chicago, Atlanta, New York, and others, as well as cities in Asia and Europe). Many US cities have developed heat vulnerability maps and action plans for mitigating heat effects in lower-income neighborhoods. Federal agencies such as the EPA Heat Islands and Equity, the CDC and NIH Heat Equity have long promoted urban heat/health equity to address problems in economically disadvantaged and racially segregated neighborhoods. In line 26, the authors cite several redlining studies on heat, but again it is not acceptable to ignore the large body of research on contemporary inequalities that underlie both the concept and methods of their study, including in their own city, Los Angeles, as well as very similar findings about neighborhood inequality. Some of the prior research shows differential impacts on heat mortality and morbidity, which this study does not do.

RESPONSE: We thank the reviewer for bringing up this important point. In the revised manuscript, we have removed the second line of introduction as recommended and updated the introduction to more comprehensively discuss the existing work on contemporary inequalities in heat hazard, exposure, and risk within cities as written below (Citations in the updated manuscript): *“Over the last few decades, research has unequivocally identified cities as distinctly warmer entities compared to their non-urban surroundings. A large body of literature has looked at intra-urban thermal landscapes and their associations with present-day population distributions. While climate change exposes rapidly growing urban populations to enhanced heat hazards, the inherent heterogeneity of cities further exacerbates the inequitable distribution of its worst impacts. For many cities, areas with poorer and predominantly non-white populations tend to be warmer. These disparities in heat hazard have been found to also lead to unequal mortality/morbidity outcomes, making them an important policy focus for equitable urban planning.”* (Please see Lines 2-9, references 1-17)

2. The manuscript would benefit from a brief section on LA climate, vegetation, and socioeconomics to establish a relevant context for the study. It is a rather unique combination of desert and coastal characteristics that differ from most other large US cities, so generalizability to other places might be questioned. The map figures are not adequately marked to give readers a sense of the expanse or “neighborhood” composition of the urban area. Sometimes the authors give interesting examples of particular places, but they are not easy to see on the maps. Figure 1 or Figure 2 might be a good place to orient readers with more prominent indications of places referred to in the text. Figure 4 is very hard to follow along with the text because it is attempting to depict relationships among three or four important variables. I do not see any red circles in Figure 4a. I recommend reviewing this section with clarity in mind.

RESPONSE: Our apologies for the oversight of not properly describing LA’s climate in the manuscript and thank you for suggesting that. We have now added a section on LA climatology and socio-economics in paragraph 3 of introduction (Lines 30-51 and also pasted below, with citations in the updated manuscript) and acknowledged in the discussion section that the findings of LA may not apply to other US cities due to its unique geography (Lines 298-300). Regarding the local neighborhoods mentioned as examples in the text, we think that labelling those on any figure may mislead a reader into assuming that our focus is solely on those particular places, when we only intend for them to serve as examples. In figure 4, there was indeed no red circle present, we have also removed any mention of the red circle from figure caption or the text.

“In this study, we focus on the Los Angeles (LA) region, home to nearly 25 million residents, which features diverse microclimates and ecological zones shaped by

complex topography and coastal influences. This leads to an uneven distribution of extreme heat events, such as heatwaves. The atmospheric circulation patterns that lead to most of Southern California's severe heatwaves are similar, typically involving a strong high-pressure system over the Pacific Northwest, which creates a significant surface pressure gradient between a high-pressure area over the Great Plains and a thermal low off the southwest coast of Los Angeles. The summer climate is largely influenced by a semi-permanent high-pressure system over the North Pacific Ocean, which deflects storm tracks and limits precipitation. The coastal upwelling of moist cold air due to dominant northwest winds during the summer leads to the formation of marine stratus clouds that cool daytime temperatures and warm nighttime temperatures. However, urbanization has been shown to slowly reduce fog and marine stratus by causing increased nighttime warming. These patterns are further modified by local winds, such as sea breezes and the occasional hot and dry Santa Ana winds during the Fall season. Heatwaves can occur when these coastal processes are disrupted, leading to higher temperatures. Over the last few decades, LA has witnessed a gradual increase in the frequency, intensity, and persistence of heatwaves posing a significant challenge to its citizens and policymakers. Additionally, LA was also subject to redlining in the past. Figure 1a, for example, displays the historically marked census tracts of LA (details in the Methods section). While the discriminatory practice was outlawed in 1968 with the passage of the Fair Housing Act, its effects still linger and continue to shape patterns of inequality in many cities. Figures 1b and 1c, for example, illustrate how green cover is drastically different between an A- and D-grade neighborhood even today. Furthermore, significant segregation is still present across historically redlined and non-redlined neighborhoods, especially in the White and Hispanic populations (Figure 1d).”

3. The definition of hysteresis in lines 164-166 is conveyed in technical terms that obscure the important substantive finding. The authors would do well to explain the specific process in a more straightforward way.

RESPONSE: We have now enhanced the explanation of the hysteresis phenomenon in a much clearer way while also preserving the technical description of the phenomenon (Lines 181-188): “Shifting our focus away from the linear correlation, we find a clear hysteresis or looping pattern when we group the data by seasons and time of day (Figure 2b). If we consider the daily maximum temperatures corresponding to afternoon hours (shown in a solid line), the thermal disparity, $\Delta LST(\text{HOLC, income})$, increases from winter to summer as the mean LST increases. Additionally, although the mean LST in spring and fall are very similar, the respective $\Delta LST(\text{HOLC, income})$ values are distinctly different. Essentially, this indicates that the seasonal variation of $\Delta LST(\text{HOLC, income})$ as a function of mean LST exhibits a time-dependent response between the forcing (i.e., mean temperature of a given day) and its effects on the system (i.e.,

thermal disparity, $\Delta LST(HOLC, income)$, on the given day). We find that this hysteresis pattern is primarily driven by variations in vegetation activity, with differences in photosynthesis and evapotranspiration cooling between spring greening and fall senescence (Supplementary Figure 8)."

4. ECOSTRESS is a welcome advance in measuring land surface temperatures. As the authors say in line 162, it is necessary to consider some other weather variables to estimate heat stress on humans. Tair at a 1 km scale is one way to do this, but it is still a rather crude estimate of microclimates. It should be acknowledged that other studies have used the Weather Research Forecasting Model (WRF) and even individually-experienced temperature sensors to improve upon air temperature measurements because air is more closely related to human stress than LST.

RESPONSE: Yes, we have improved the discussion in section 2.4 Beyond Exposure where we discuss the limitation of LST in estimating exposure, and utility of T_{air} (air temperature) as a more appropriate metric (Lines 200-207).

5. It is standard to use PCA to construct heat vulnerability indexes. The authors have included some atypical variables in their analysis, which is good, but they don't explain or label the factors. If I understand correctly, only the first factor that explains 35% of the variance is used in the vulnerability index. More discussion of what this analysis shows about the social sources of heat vulnerability would be helpful. I don't see an explanation for the choice to omit the other two factors that increase the explained variance in the data. Also, please give clear color guidance in Supplemental Figure 11 for the significance and direction of factor loadings.

RESPONSE: In response to the reviewer's request for clarification, we provide the following interpretations of the first three principal components (PCs) from our analysis which has now been added to the latest version of our manuscript (Lines 226-240).

"To obtain a single condensed metric of heat vulnerability, we performed a principal component analysis (PCA) on the identified variables. Through an in-depth exploration of variable loadings, we find that the first PC explains more than one-third of the observed variance (eigenvalue: 10.0, explained variance: ~35%). PC1 captures a multidimensional sensitivity to heat with strong and positive correlations with health variables such as obesity, diabetes, and asthma, as well as moderate positive correlations with socioeconomic factors such as extreme poverty, under-education, energy deficit households, predominantly outdoor workers, and overcrowded and inadequate housing conditions. PC2 (eigenvalue: 4.0, explained variance: ~13%) has strong correlations with individuals who live alone, renters, and those lacking access to personal vehicles or the ability to drive to work, reflecting an additional sensitivity to heat

driven by social isolation and harsh working conditions. Lastly, PC3 (eigenvalue: 2.5, explained variance: ~8%) captures a segment of the population that is elderly, lives alone, and experiences adverse health conditions such as diabetes and stroke (further details in Supplementary Figure 11b). Each PC is by definition orthogonal to the others, and thus they are best treated as separate dimensions of vulnerability. In the remainder of this work, we focus on the first PC only, given that it is the most comprehensive, and refer to it as the 'vulnerability score'."

To provide additional context for our decision to retain only PC1 in the vulnerability index: In our preliminary analysis, we explored the first few principal components (PCs) and examined their spatial distributions. Given that each PC is orthogonal to the others, they represent separate dimensions of heat vulnerability. Importantly, each PC reflects contributions from all the input variables, with factor loadings detailed in Supplementary Figure 11. Although we considered combining multiple PCs into a single vulnerability score, we ultimately chose not to. While PC2 and PC3 did increase the explained variance slightly, the gain was not substantial enough to outweigh the loss of interpretability that comes with merging dimensions that capture different aspects of vulnerability. As such, we proceeded with PC1 as the primary vulnerability score. We have also updated Supplementary Figure 11 to provide descriptions for each variable. The color gradient has been clarified: red indicates a positive loading, blue a negative loading, and the shading reflects the strength of each relationship.

6. The discussion of “access” to air conditioning in lines 231-238 is somewhat confusing. Does access mean having the equipment in the home, choosing to use it or not, or not using it because it is unaffordable? Considering other research on indoor heat deaths and social surveys of customers, it is likely that residents in low-income areas don't use air conditioning because they can't afford it. In Supplemental Figures 11 and 15, it looks like low usage areas are in downtown LA where African Americans and Latinos live.

RESPONSE: Our data relies on estimates of “Probability of access to AC” from Romitti et al (2022) that was derived from various demographic factors, climate conditions, and geographic differences. This is more closely related to probability of ownership of ACs. In most regions of the US where AC penetrations tend to be very high, this correlation is acceptable, but California has lower penetrations than most US states because of the coastal influence, and this relationship breaks down. As the reviewer pointed out, this doesn't necessarily imply usage (for example, affluent houses in cooler micro-climates such as LA coasts and mountains where they can afford AC but need not use it, or poor houses where despite an AC unit, the electricity bill might be untenable). Therefore, we improve upon this estimate by assessing the patterns in their electricity consumption as a function of mean temperature of a given day using smart-meter records of 200,000

homes in Southern California over the years 2015-2016 provided by local utility Southern California Edison (SCE). As a result, our estimates capture the actual AC usage better (Chen et al, 2019; 2020). We have explained this better now in Section 4.6 Estimates of Air Conditioning access and usage (Lines 356-366).

7. This brings me to my final comment that refers to Supplemental Figures 13 and 14. These figures show that African Americans are as highly exposed to heat risks as Hispanics in LA. I believe the data indicates that lines 266-267 in the Discussion should be amended accordingly.

RESPONSE: Thanks for bringing this to our attention. We have amended the abstract and discussion section (Line 302) accordingly.

References:

- Romitti, Y., Sue Wing, I., Spangler, K. R. & Wellenius, G. A. Inequality in the availability of residential air conditioning across 115 us metropolitan areas. PNAS nexus 1, pgac210 (2022).
- Chen, M., Sanders, K. T. & Ban-Weiss, G. A. A new method utilizing smart meter data for identifying the existence of air conditioning in residential homes. Environ. Res. Lett. 14, 094004 (2019).
- Chen, M., Ban-Weiss, G. A. & Sanders, K. T. Utilizing smart-meter data to project impacts of urban warming on residential electricity use for vulnerable populations in southern california. Environ. Res. Lett. 15, 064001 (2020).

Reviewer #2:

This article is an extremely important contribution to the literature about urban heat islands. It elucidates several important drivers of inequitable vulnerabilities to climate change, in particular dissecting the relationship between historical redlining and current economic and racial disparities. The outcomes are well-articulated in the introduction and discussion. It should be published ASAP.

The robust statistical analysis and referencing of prior research and source datasets are exceptional. While I do not conduct this type of quantitative analysis in my own research, with a bit of effort, I could follow the analysis, with two exceptions, as explained below.

1. First, the explanation of how the first PC was used to create the vulnerability score should be strengthened. What exactly does the scale of -6 to 6 represent? And what does the red and blue color coding represent? The first question should be answered more clearly at the end of the second paragraph in Section 2.4. The second question could be answered more clearly in the caption in Supplementary Figure 11b.

RESPONSE: We appreciate the reviewer's suggestion and have made revisions to address both points more clearly. As part of our preliminary analysis, we considered the first 3 principal components (PC's) and analyzed their spatial distributions. Each PC is orthogonal to all others, and should be treated as a separate dimension. We considered combining the PCs into a single 'vulnerability score', but the additions in explained variance of PC2 and PC3 were not large enough to justify the loss of interpretability due to the combination of different dimensions, so we retained and performed the rest of our analysis only with PC1 as the vulnerability score. We have strengthened this explanation in the text now (Lines 226-240).

Regarding the scale of -6 to 6: This scale represents the range of PC1 scores, with each score indicating a specific level of vulnerability relative to the overall data distribution. A higher score (closer to 6) corresponds to greater vulnerability, while a lower score (closer to -6) indicates less vulnerability. We have updated the figure 6 caption to clarify that this range reflects standardized values based on the contributions of the variables loading onto PC1.

As for the red and blue color coding in Supplementary Figure 11b, we have revised the caption to explain that red indicates a positive loading, meaning the variable contributes to increased vulnerability, while blue represents a negative loading, indicating that the variable mitigates vulnerability. The degree of shading reflects the strength of each variable's contribution.

2. Second, a more minor addition I recommend, also in Section 2.4, is to explain the difference between LST and near-surface air temperature. In particular, the only place I could find an indication of the height above ground at which the air temperature was taken is in the caption for Supplementary Figure 10. It would be helpful to readers if this distance were noted in the description of the dataset in the narrative in Section 2.4.

RESPONSE: Thank you for pointing that out. We have now added the following clarification regarding near surface air temperature to Section 2.4 which should be helpful for the readers: Estimating heat exposure requires consideration of variables beyond surface temperature, such as near-surface air temperature (T_{air} , which is typically measured at 2 meters above ground level), wind speed, solar irradiance, and humidity (Lines 201-204).

3. Line 32, sentence starting “Further...” needs editing for clarity, as it appears to be an incomplete sentence

RESPONSE: We have rewritten that sentence for improved clarity.

4. Line 190, “maybe” should be “may be”

RESPONSE: Thank you, we have corrected that now.

Reviewer #3:

The study aims to disentangle the effects of historic redlining and current income inequality on urban heat disparities. The results indicate that present-day income inequality is a stronger indicator of heat burden than historic redlining. While the research addresses an important topic—*intra-urban inequities*—there are several critical issues that need to be addressed to strengthen the manuscript.

Major comments:

1. This study focuses solely on one city, Los Angeles. To attract more attention, the narrative should be framed from a broader perspective. For example, it could include major cities across the United States or even globally. The implication of this study is limited to a single city.

RESPONSE: At this stage, it is indeed a single-city focussed paper, since the signals we are looking at are small and is influenced by many factors (LA's climate, coastal influence, specific history as detailed in paragraph 3 of introduction) that may not be generalizable to all cities in the U.S. Moreover, redlining is not a global phenomenon and the urban population distributions in the U.S. is quite unlike that in other countries. However, this study provides a methodological framework to better separating out historical versus present inequalities for heat disparity studies over other cities. To avoid miscommunication of the scope of our study, we have updated the title to explicitly include LA.

2. Lines 3-4: Previous studies have extensively analyzed *intra-urban* inequalities. It is necessary to further expand the background information and include relevant citations. Additionally, it is important to highlight the limitations of the previous studies and the innovative aspects of this research in the Abstract and Introduction.

RESPONSE: Yes, we have now rewritten the introduction and abstract to reflect that better (Citations in the updated manuscript): *“Over the last few decades, research has unequivocally identified cities as distinctly warmer entities compared to their non-urban surroundings. A large body of literature has looked at intra-urban thermal landscapes and their associations with present-day population distributions. While climate change exposes rapidly growing urban populations to enhanced heat hazards, the inherent heterogeneity of cities further exacerbates the inequitable distribution of its worst impacts. For many cities, areas with poorer and predominantly non-white populations tend to be warmer. These disparities in heat hazard have been found to also lead to unequal mortality/morbidity outcomes, making them an important policy focus for equitable urban planning.”* (Lines 2-9, references 1-17)

The main limitations of previous studies and our key innovative use of ECOSTRESS is acknowledged in lines 52-61: *“Most previous investigations into the intra-urban and intra-HOLC heat inequalities have relied on Land Surface Temperatures (LST) estimated for specific times of day from satellites in sun-synchronous orbits (e.g., Landsat at 10:00 am). Here, we overcome this limitation in temporal resolution by using data from the ECOSTRESS (ECOsystem Spaceborne Thermal Radiometer Experiment on Space Station) thermal instrument onboard the International Space Station (ISS) whose precessing orbit allows observations at different times during the day, providing comprehensive temporal coverage. By leveraging ISS-borne LST observations, we aim to discern the diurnal and seasonal patterns that may have been overlooked in previous studies, providing a more detailed perspective on the temporal dynamics of heat inequalities and their potential implications for vulnerable and historically underserved communities.”*

3. Lines 38-41 and Lines 283-292: What is the spatial coverage of ECOSTRESS observations? Can it be used for global or U.S.-wide studies? Why was Los Angeles chosen as the study subject? Moreover, this study focuses only on cloud-free land surface temperatures. Considering the influence of clouds on temperature could impact the study results. For instance, higher temperatures promote precipitation and cloud cover, thus reducing incoming solar radiation and mitigating urban warming. This limitation should be acknowledged in the article.

RESPONSE: While ECOSTRESS indeed has a global coverage, we chose to focus this research on a single city - LA for now because it is a large enough metropolitan city to have high intra-urban climatological, topographical, and ecological diversity and a history of redlining (Paragraph 3 of introduction, Lines 30-51). Moreover, beyond the fine-scale socio-economic data, we have information about air conditioning usage here through our collaborator and co-author Kelly Sander’s research group at USC. Although redlining is not a global phenomenon, this study provides a methodological framework to better separating out historical versus present inequalities for heat disparity over other cities. Considering the influence of clouds on temperature is important and as they obscure surface temperature observations, it is a limitation of LST based analysis indeed. We have acknowledged that in Line 200-207.

4. Land surface temperature does not accurately reflect human thermal sensation. It is necessary to supplement the study with surface air temperature data from meteorological stations. Moreover, the temporal resolution and frequency of the data are not clearly described in the manuscript. It is not clear what data has been used to derive the average LSTs.

RESPONSE: The reason we use LST data is because we do not have near-surface air temperature data at the necessary spatial resolution (redlined neighborhood scale) for

this study. However, we do use a gridded air temperature data to contextualize some of our LST based results and discuss those in lines 200-216 and Supplementary figure 10.

Detailed explanation of our data source is described in the Methods section (Section 4.2, Lines 320-328) at the end of the paper as per the journal's guideline. Our primary dataset is the standard Land Surface Temperature (LST) product from ECOSTRESS with a spatial resolution of 70x70 meters (at nadir) acquired over the LA region since mid-2018. ECOSTRESS, or the ECOsystem Spaceborne Thermal Radiometer Experiment on Space Station, was launched to the International Space Station in June 2018, and it currently provides the highest combined spatial (70m at nadir), spectral (3/5 bands), and temporal (3-5 days) resolution thermal infrared data from space with observations available over the diurnal cycle. As ECOSTRESS is not a polar orbiting satellite, but an instrument mounted on the International Space Station, it doesn't have a regular frequency of revisit over any spot. However, the irregular overpass time is precisely why we are able to leverage its irregular revisit time to obtain LST observations at any time of day for this study.

Minor comments:

1. Line 83 and Lines 102-103: In addition to seasonal differences, the reasons for daytime and nighttime variations should also be specifically discussed and analyzed.

RESPONSE: The diurnal variations are discussed and analyzed in Section 2.1 as well as Section 2.2

2. Lines 127-129: During winter, why are temperatures higher in high-income areas compared to low-income areas? Is it due to the effect of anthropogenic heat?

RESPONSE: This is likely an effect of thermal regulation provided in the form of evapotranspiration by the local vegetation in the affluent neighborhoods that keeps the winter nights warmer than the poorer neighborhoods (Lines 144-146). This was also observed and discussed in details in another study (Yin, et al. 2023) by our co-author, Dr. Yi Yin.

3. Lines 263-264: How is a heatwave event defined in the study?

RESPONSE: We use hotter than 95th percentile of the usual climatology as the heatwave threshold in this study. However, the focus isn't a heatwave event here but any extremely hot day. We have edited this sentence to the following for improved clarity: We find that the thermal disparities escalate to up to 5-7°C during extreme heat days such as heatwaves and exhibit a seasonal hysteresis pattern driven by the springtime greening and falltime senescence of local vegetation (Lines 296-298).

4. The research indicates that present-day income inequality is a stronger indicator of heat burden than historic redlining. This content is not reflected in the title.

RESPONSE: This is indeed a valuable suggestion. We have updated the title to appropriately capture our most notable finding now: “Present-day inequities in Los Angeles have a stronger influence on intra-urban heat disparities than historic redlining”.

5. The abstract should briefly introduce the background of historic redlining. Or at the beginning of the introduction.

RESPONSE: Paragraph 2 of the introduction (Lines 15-29) and the Methods Section 4.1 (Lines 311-318) describes the background historic redlining and Line 1 and 2 of Abstract summarizes that.

References:

- Yin, Y., He, L., Wennberg, P. O., & Frankenberg, C. (2023). Unequal exposure to heatwaves in Los Angeles: Impact of uneven green spaces. *Science Advances*, 9(17), eade8501.

Reviewer #1 (Remarks to the Author):

Thank you to the authors for addressing the reviewers' commentary. Their efforts have improved the manuscript. Reviewer 1's reactions to revisions made in response to my comments:

1. Thank you for removing the misleading sentence in the first paragraph. Although the number of citations added does not fully reflect the large literature on intra-urban heat inequalities, it makes the necessary point for this manuscript.

Response: Thank you.

2. The new description of the LA climate and geography is not exactly what I had in mind. The complexities of the Southern California climate zones might be better appreciated by referring to the unusual close proximity of Koppen classifications; i.e., something like lines 278-279 would be beneficial in the introduction. Regarding comments about the locations of places on the maps, a larger point is that we should have an exact definition of the boundaries of the study area. The "Los Angeles region" is too vague – is it an MSA or some other construction that you are using? The use of terms like city and neighborhoods throughout the text deserves some clearer context because there are many cities in this region. Figure 2 is an example – for readers who are not familiar with SoCal, there is no orientation to where the city of LA is, let alone central LA, urban canyons, and shipping dock. Is a popout map an option? Figure 3 gives some markers about where places are located.

Response: I see, thank you for that clarification, we have now rewritten the geographical and climatological background of LA with more of a focus on Koppen Geiger classification (Lines 30-35). We are using the census tracts within LA county and have clearly specified that in Line 30. While a pop-out map is not an option at this stage, figure 2 is indeed a good opportunity to label some major landmarks to properly situate the reader. Thank you for that suggestion. We have now labeled Figure 2 accordingly (please see below).

3, 4. Thank you for the clarifications.

5. I might have made a different decision about how to use the PCA. Collectively, I think we are learning this method is not the best way to differentiate the different types of human vulnerability. Factor 1 points to a broad constellation of vulnerable individuals but it is so general that it would not provide much guidance on protections. Nevertheless, that is not the point of this study and we don't have anything better as a summary measure.

Response: We appreciate the reviewer's observation about the first principal component (PC1). PC1 here (as well as all the other PCs) is an aggregate measure of vulnerability and the various contributions to its definition are shown in SI Figure 11b. Owing to the non-Gaussian nature of the variables that describe the vulnerability, we cannot expect a simpler description of PC1. As the reviewer also notes, our aim in using PCA was solely to create a low-dimensional, aggregate measure of vulnerability to compare with other indices (income and redlining score). In this regard, the use of PCA meets the intended purpose.

6, 7. Thank you for the clarifications.